# Regeneration and Recovery after Acetaminophen Hepatotoxicity

**Bharat Bhushan** [1,2,*] **and Udayan Apte** [3]

1 Department of Pathology, University of Pittsburgh School of Medicine, Pittsburgh, PA 15261, USA
2 Pittsburgh Liver Research Center, University of Pittsburgh Medical Center, University of Pittsburgh School of Medicine, Pittsburgh, PA 15261, USA
3 Department of Pharmacology, Toxicology and Therapeutics, University of Kansas Medical Center, Kansas City, KS 66160, USA; uapte@kumc.edu
* Correspondence: bhb14@pitt.edu; Tel.: +1-412-648-2021

**Abstract:** Liver regeneration is a compensatory response to tissue injury and loss. It is known that liver regeneration plays a crucial role in recovery following acetaminophen (APAP)-induced hepatotoxicity, which is the major cause of acute liver failure (ALF) in the US. Regeneration increases proportional to the extent of liver injury upon APAP overdose, ultimately leading to regression of injury and spontaneous recovery in most cases. However, severe APAP overdose results in impaired liver regeneration and unchecked progression of liver injury, leading to failed recovery and mortality. Intercommunication between various cell types in the liver is important for effective regenerative response following APAP hepatotoxicity. Various non-parenchymal cells such macrophages, stellate cells, and endothelial cells produce mediators crucial for proliferation of hepatocytes. Liver regeneration is orchestrated by synchronized actions of several proliferative signaling pathways involving numerous kinases, nuclear receptors, transcription factors, transcriptional co-activators, which are activated by cytokines, growth factors, and endobiotics. Overt activation of anti-proliferative signaling pathways causes cell-cycle arrest and impaired liver regeneration after severe APAP overdose. Stimulating liver regeneration by activating proliferating signaling and suppressing anti-proliferative signaling in liver can prove to be important in developing novel therapeutics for APAP-induced ALF.

**Keywords:** hepatotoxicity; liver regeneration; proliferation; drug-induced liver injury; Cyclin D1; p21; TGFβ1; β-catenin





## 1. Introduction

The liver has an extraordinary capacity to regenerate upon surgical resection, infection, and toxicant- or chemical-induced liver injury. The liver is the only organ that restores exactly to its original mass even after two-thirds of tissue loss [1]. Each cell type in the liver including hepatocytes, cholangiocytes, hepatic stellate cells, and endothelial cells usually proliferate to replace their own cell type ultimately attaining original liver mass and function. The phenomenon that regulates liver size in normal physiology and especially during compensatory regeneration has been termed as the hepatostat [1,2]. Liver regeneration is orchestrated by synchronized actions of several proliferative signaling pathways involving numerous kinases, nuclear receptors, transcription factors, transcriptional co-activators, which are governed by cytokines, growth factors, endobiotics such as bile acids, and hormones [3]. Typical regenerative response of healthy liver is very robust, and the elimination of any one signaling pathway does not usually have a major impact on overall outcome due to redundancy of the signaling mechanisms [1].

Extensive studies have shown that liver regeneration plays a crucial role in recovery after acetaminophen (APAP) overdose, a major clinical problem [4–9]. APAP is one of the most used over-the-counter analgesics and is considered very safe at therapeutic doses. However, APAP overdose results in acute liver injury involving hepatocyte cell death and centrilobular necrosis, which in severe cases leads to acute liver failure (ALF). APAP

overdose is the topmost cause of ALF in the United States and the Western world [10]. APAP-induced hepatocyte death and centrilobular necrosis is initiated by its toxic metabolite NAPQI (N-acetyl-p-benzoquinone imine), which is normally excreted after conjugation with cellular glutathione (GSH). Excess NAPQI accumulated after APAP overdose depletes cellular GSH stores and forms adducts with cellular proteins (majorly with mitochondrial proteins), ultimately leading to mitochondrial damage, release of cell death mediators from mitochondria, and necrosis [11,12]. Interestingly, most of the APAP overdose patients recover spontaneously due to robust compensatory regenerative response. During this compensatory regenerative response, dead hepatocytes are replaced by newly formed cells, originating from viable hepatocytes surrounding the necrotic zones leading to restoration of normal liver mass and function. However, in severe cases, spontaneous recovery does not occur because of delayed and inhibited liver regeneration leading to mortality. Numerous studies have correlated robust liver regenerative response with better outcome in APAP-induced ALF patients [13,14]. Thus, timely stimulation of liver regeneration is a potential therapeutic option for APAP-induced ALF, and understanding the mechanisms of liver regeneration is important for developing potential therapeutic targets. Regenerative therapies can be especially beneficial considering liver injury is already established in most of the patients by the time they seek medical attention and is difficult to manipulate. This is evident from the fact that N-acetyl cysteine (NAC), the only pharmacological therapy of APAP-induced ALF, which is based on intervening early stages of liver injury, is not effective in late presenting patients [15]. In contrast, regenerative interventions can be potentially applied even at a later stage.

In the past, most of the studies to understand the mechanisms of liver regeneration have been focused on a partial hepatectomy model, which involves resection of healthy liver [16]. However, APAP hepatotoxicity is complicated by presence of massive cell death and subsequent persistent inflammation. These processes and associated signaling mediators intricately regulate the liver regeneration response after APAP hepatotoxicity. Recent studies have shown that liver regeneration response and the underlying mechanisms are very different in this toxic environment compared to the healthy liver [17]. Further, other underlying pathological conditions such as alcoholic or non-alcoholic fatty liver disease can also impact liver regeneration response. Liver regeneration is much more synchronized after partial hepatectomy of healthy liver compared to regeneration in response to APAP hepatotoxicity, where several injury related factors can actively impede proliferation [17]. Thus, in recent years, studies have focused on delineating liver regeneration mechanisms specifically in an APAP-induced liver injury model. The mechanisms that promote or inhibit liver regeneration specifically after APAP hepatotoxicity, their dose–response characteristics, and the role of various hepatic cell types in the liver regeneration response are discussed in this review. Additionally, factors to consider while designing studies to investigate liver regeneration following APAP hepatotoxicity are also discussed.

## 2. Dose–Response Characteristics of Liver Regeneration after APAP Hepatotoxicity

Hepatocyte proliferation and liver regeneration occur as a compensatory response to exposure of any injurious chemical toxicants [18]. These include toxicants that cause injury to different zones of liver, including centrilobular hepatotoxicants (such as carbon tetrachloride, thioacetamide, and APAP) and periportal hepatotoxicants (such as allyl alcohol) [9,18–21]. Detailed dose–response characteristics of liver regeneration following these toxicant-induced liver damage have been extensively studied and have been reviewed earlier [18]. These studies have established that liver regeneration increases proportional to the liver injury upon increasing the dose of toxicant [18,20]. There is also a progressive delay in liver regeneration response with increasing the dose. However, the incremental liver regeneration response more than offsets the delay in regenerative response up to a certain dose, ultimately inducing regression of liver injury and recovery. Thus, up to a threshold dose, stimulation of liver regeneration occurs proportionate to the extent of injury. Beyond this threshold dose, there is a sharp decline and delay in liver regeneration response upon

increasing the dose further. This results in unchecked progression of liver injury leading to failed recovery and mortality [9,18]. For instance, this dose–response relationship of liver regeneration has been well characterized for thioacetamide, utilizing several different doses (50, 150, 300, and 600 mg/kg) over a time-course of 0–96 h in rats. Liver regeneration increases proportionate to liver injury up to 300 mg/kg dose of thioacetamide leading to complete recovery. However, liver regeneration is severely compromised and delayed at a 600 mg/kg dose leading to significant mortality [20]. Further, if proliferation is blocked by administering anti-mitotic agents (such as colchicine) after doses of toxicants that normally result in robust liver regeneration and spontaneous recovery, it also results in progression of injury and failed recovery [18,22]. These studies further emphasize the importance of liver regeneration for recovery and regression of injury.

Studies utilizing incremental doses of APAP in mice have shown similar dose–response characteristics of liver regeneration after APAP hepatotoxicity [9]. A moderately toxic dose of APAP (300 mg/kg) in mice causes extensive liver injury but also timely and robust compensatory liver regeneration, ultimately leading to regression of injury, spontaneous recovery, and survival. In contrast, severe APAP overdose (600 mg/kg) in mice results in comparable initial liver injury but disproportionately impaired and delayed liver regeneration, resulting in unchecked progression of injury, failed recovery, and significant mortality [9]. Other doses of APAP such as 350, 450, and 525 mg/kg of APAP have also been utilized for investigating liver regeneration [23]. However, a more comprehensive dose–response relationship of liver regeneration in the APAP model, including several different doses of APAP, and all temporal phases of APAP hepatotoxicity and compensatory regeneration still needs to be documented. Lastly, the mechanisms involved in liver injury and compensatory regeneration vary greatly with the dose of APAP. Thus, it is important to study the mechanisms of liver injury and regeneration utilizing multiple doses, especially severely toxic doses of APAP, which are not well studied as the majority of previously published studies utilize only moderately toxic doses of APAP, where animals regenerate spontaneously. Understanding these mechanisms at severely toxic doses of APAP is also clinically relevant to mimic pathophysiology of APAP-induced ALF patients who fail to recover spontaneously and require liver transplantation. Thus, it is important to consider if doses utilized in a study investigating liver regeneration are relevant for exposure observed in APAP overdose patients, especially in those who do not recover spontaneously. Overall, the dose–response relationship of liver regeneration following APAP hepatotoxicity is very important to consider for any study design focusing on understanding the mechanisms of liver regeneration or developing regenerative therapeutics for APAP-induced ALF.

## 3. Multiple Proliferative Signaling Mediators Contribute to Orchestrate Liver Regeneration Following APAP Hepatotoxicity

Liver regeneration is orchestrated by complex interplay of several cytokines, mitogens, and other proliferative signaling pathways following APAP hepatotoxicity [1]. While cytokines (such as tumor necrosis factor alpha: TNF-$\alpha$ and Interleukin 6: IL-6) are considered auxiliary mitogens for liver as they alone do not induce hepatocyte proliferation in vivo or in vitro, growth factors such as hepatocyte growth factor (HGF) and epidermal growth factor (EGF) receptor ligands (e.g., EGF and transforming growth factor alpha: TGF-$\alpha$) are considered primary mitogens for liver as they can alone induce hepatocyte proliferation in vivo or in vitro even in serum free chemically defined medium [1,3]. Further, the EGF receptor (EGFR) and HGF receptor (i.e., c-MET) are the only known cell membrane receptors whose signaling disruption in combination results in complete elimination of liver regeneration response after partial hepatectomy, highlighting importance of these receptor tyrosine kinases [24]. Elimination of any other upstream signaling pathway causes only delay in liver regeneration response after partial hepatectomy, but the liver eventually regenerates to attain hepatostat. Temporal and dose-dependent dynamics of activation of growth factors, cytokines, and other important pro-regenerative signaling have also been

comprehensively investigated in the context of APAP-induced liver injury as discussed in the following part of this section [9].

Both c-MET and EGFR are robustly activated dose-dependently after APAP overdose in mice and might play an important role in liver regeneration following APAP hepatotoxicity [9]. Interestingly, these receptors are activated very early following APAP overdose in mice, even prior to any observable liver injury. EGFR is activated within the first 30 min of APAP administration in mice and remains activated in sustained manner even during the recovery phase (up to 96 h after APAP) [8,9]. EGFR activation during the recovery phase might be crucial for driving regenerative response as late administration of EGFR inhibitor (after liver injury is already established) not only impairs liver regeneration but also leads to failed recovery and significant mortality after a moderately toxic dose of APAP (300 mg/kg) that normally culminates in spontaneous recovery in mice [8]. Paradoxically, EGFR activation during the injury initiation phase might be linked to cell death signaling as early EGFR inhibition (prior to injury development) results in almost complete attenuation of liver injury following APAP overdose in mice, eliminating any need for compensatory liver regeneration [8]. c-MET is also activated very early (within 3 h) after APAP overdose in mice, but its causal role in liver regeneration in the APAP-induced ALF model has not yet explored [9]. Further, the role of both EGFR and c-MET in liver injury or regeneration after APAP overdose needs to be established using specific genetic deletion approaches considering the potential off-target effects of chemical inhibition.

The role of cytokine signaling (TNF-$\alpha$/NF$\kappa$B and IL-6/STAT3) is relatively more extensively studied in the APAP model utilizing transgenic mice. Both TNF-$\alpha$ and IL-6 expression levels in liver increase after APAP overdose in mice [9,25]. Further, the deletion of TNF receptor 1 (TNF-R1) and IL6 in mice results in impaired liver regeneration after APAP hepatotoxicity [6,26,27]. Moreover, the administration of IL-6 in IL-6 knock-out mice results in restoration of the liver regeneration response [6]. Interestingly in our previous study, TNF$\alpha$/NF$\kappa$B signaling activation, downstream nuclear translocation of NF$\kappa$B, and its binding to the promoter of core cell cycle genes (such as Cyclin D1, which governs entry into cell cycle) were greater after moderate APAP overdose, which was accompanied by robust liver regeneration response. However, TNF$\alpha$/NF$\kappa$B signaling activation was remarkably inhibited after severe APAP overdose which correlated with impaired liver regeneration [9]. In contrast, IL-6/STAT-3 signaling activation was dose-dependently higher after severe APAP overdose in mice [9]. This suggests dose-dependent differential role of these cytokines in liver regeneration after APAP hepatotoxicity, which needs further exploration.

Other than growth factors and cytokine signaling, Wnt/$\beta$-catenin signaling is also considered to be very important for hepatocyte proliferation and liver regeneration [1]. The role of the Wnt/$\beta$-catenin signaling pathway has also been studied in the APAP model of liver regeneration using transgenic and pharmacological approaches. Similar to the TNF$\alpha$/NF$\kappa$B signaling pathway, $\beta$-catenin signaling activation, nuclear translocation of $\beta$-catenin, and its binding to the Cyclin D1 promoter occurs robustly at a regenerating dose of APAP in mice but is inhibited at doses where liver regeneration is impaired [9]. B-catenin signaling activation is also correlated with higher liver regeneration and survival in ALF patients [14]. Utilization of the $\beta$-catenin deletion strategy to demonstrate its causal link with liver regeneration in the APAP model has been hampered as these mice exhibit very low expression of Cyp2e1, the main enzyme involved in metabolic activation of APAP, and thus exhibit low hepatotoxicity compared to wild-type (WT) mice [14]. However, $\beta$-catenin deletion in the liver results in impaired liver regeneration when different doses of APAP are utilized in WT and $\beta$-catenin KO mice to achieve equal liver injury (i.e., equitoxic dose strategy), and consistent overexpression of $\beta$-catenin results in significant stimulation of liver regeneration [9,14]. $\beta$-catenin signaling has also been targeted pharmacologically to develop regenerative therapy for APAP-induced ALF by using the inhibitor of glycogen synthase kinase 3 (GSK3), which is an upstream inhibitor of $\beta$-catenin. However, pharmacological activation of $\beta$-catenin signaling by inhibiting GSK3 results only in early onset

of a proliferative response after severe APAP overdose, without significantly affecting the peak regenerative response or overall outcome/survival [28]. Although several studies have established β-catenin to be a critical regulator of liver regeneration following APAP hepatotoxicity, the specific Wnt ligands that activate β-catenin and the source of these Wnt ligands remain elusive in this model.

Apart from the Wnt/β-catenin signaling pathway, the Hippo/YAP signaling pathway has been also emerged as a critical regulator of hepatocyte proliferation and liver size in recent years [1]. YAP signaling activation, which is normally associated with higher proliferation and hepatomegaly in liver, is also rapidly activated during APAP overdose in mice [29,30]. However, in the context of APAP hepatotoxicity, YAP appears to inflict the opposite effect as hepatocyte-specific YAP deletion results in faster hepatocyte proliferation and rapid recovery after APAP hepatotoxicity in mice [29]. Further, rapid recovery from APAP overdose in the hepatocyte specific YAP knockout mice is related to faster activation of the Wnt/β-catenin pathway [29]. Lastly, several endobiotics such as bile acids and hormones can also potentially regulate liver regeneration following APAP hepatotoxicity based on knowledge from the partial hepatectomy model. Indeed, signaling via bile acids can contribute to liver regeneration response following APAP hepatotoxicity as both cholic acid and FGF19 (a downstream mediator of bile acid signaling) treatment result in improved liver regeneration after APAP overdose in mice [31,32].

## 4. Signaling Mechanisms Involved in Inhibiting Liver Regeneration Following APAP Hepatotoxicity

One of the most interesting findings with regard to liver regeneration in the APAP model is that many of the proliferative signaling pathways considered very critical for liver regeneration remain highly activated even after severe APAP overdose in mice, but liver regeneration is still severely impaired at these high doses [9]. For example, primary mitogen signaling via HGF/c-MET and EGF/EGFR pathways and their downstream ERK signaling are more activated after severe APAP overdose, where liver regeneration is impaired [8,9]. Similar is the case of cytokine signaling via the IL-6/STAT-3 pathway [9]. Interestingly, the majority of the hepatocytes are still viable even after severe APAP overdose in mice, but they fail to respond to these proliferative signals [9]. This indicates the possible contribution of mediators that actively inhibit cell cycle, leading to cell cycle arrest at severe APAP overdose, where liver regeneration is impaired. Although activation of cell cycle inhibitory mechanisms is important for balanced proliferative response and effective DNA repair, overt activation of these pathways after severe APAP overdose may result in impaired liver regeneration and failed recovery. Indeed, studies have shown striking activation of key cell cycle inhibitors such as p21 and p53 after severe APAP overdose in mice [9,33]. Further, the deletion of p21 and p53 results in higher or faster liver regeneration after APAP overdose in mice [23,34]. Excessive double strand DNA damage and limited DNA repair pathways activation in peri-necrotic regions might be responsible for the activation of cell cycle arrest signaling after severe APAP overdose [33]. Clinical relevance of all these findings can be appreciated by studies showing association of hepatic DNA damage, increased expression of cell cycle inhibitors such as p21, and cell cycle arrest with impaired liver regeneration response in APAP-induced ALF patients [23,35].

Transforming growth factor beta (TGFβ) appears to be one of the important upstream signaling pathways to be involved in induction of p21 and senescence in perinecrotic areas after severe APAP overdose [23]. TGFβ signaling is activated in perinecrotic areas after APAP overdose in mice/humans correlating to p21 induction and the deletion of the *Tgfb1* gene, or treatment with the TGFβ receptor 1 inhibitor in mice decreases p21 expression and improves liver regeneration/survival [23]. Macrophages were found to be an important source of TGFβ driving anti-proliferative effects in the above study [23]. Treatment with the TGFβ1 inhibitor can be a potential therapeutic option for stimulating regeneration after APAP overdose. Apart from TGFβ, extracellular matrix signaling transduction via integrin-linked kinase (ILK) is also important for producing anti-proliferative effects on

hepatocytes, maintaining a quiescent state, and in the termination of liver regeneration following partial hepatectomy [36]. A similar role of ILK in inflicting inhibitory effects on hepatocyte proliferation and liver regeneration has also reported in the APAP overdose model as a liver-specific ILK deletion resulted in striking increase in hepatocyte proliferation disproportionate to the extent of liver injury after APAP overdose in mice [37].

## 5. Liver Regeneration after APAP Hepatotoxicity Involves Complex Interplay of Hepatic Parenchymal and Non-Parenchymal Cells

As mentioned previously in this review, each hepatic cell type usually proliferates to replace its own cell type during the process of liver regeneration [3]. However, each hepatic cell type can produce mediators which are important for proliferation of other cell types in liver [1]. Moreover, extrahepatic tissues can also produce some of the important growth factors involved in liver regeneration. For instance, Brunner glands of the duodenum are the major source of EGF exposed to liver via portal circulation, which is an important primary mitogen for hepatocytes. Stellate cells are the major source of HGF in liver, which is another major primary mitogen for hepatocytes. Stellate cell depletion or treatment with stellate cell-derived conditioned medium produces the opposite effect of inhibited or improved hepatocyte proliferation/liver regeneration, respectively, after APAP hepatotoxicity in mice [38,39]. Similarly, endothelial cell proliferation is not only important for restoring hepatic vasculature but also for hepatocyte proliferation as vascular endothelial growth factor (VEGF)-stimulated endothelial cells also produce HGF. Indeed, treatment with the VEGF inhibitor or deletion of VEGF receptor 1 (Vegfr-1) in mice results in impaired hepatocyte proliferation after APAP hepatotoxicity, and treatment with recombinant VEGF results in increased hepatocyte proliferation following APAP hepatotoxicity [5,40,41]. Lastly, macrophages recruited to centrilobular necrotic areas after APAP hepatotoxicity are not only important for removal of cell debris to house newly formed hepatocytes, but they are known to produce both cytokines (TNF-$\alpha$, and IL-6) and mitogens (HGF and TGF-$\alpha$) important for hepatocyte proliferation. Liver resident macrophages (Kupffer cells) can promote proliferative signaling in hepatocytes via induction of chemokine receptor CXCR2 [42]. The therapeutic potential of targeting macrophages has been also demonstrated by a recent study where cell-based therapy involving administration of alternatively activated macrophages promoted hepatocyte proliferation and resolution of APAP-induced liver injury [43]. Thus, inter-communication between different cell types in liver is important for an effective regenerative response following APAP hepatotoxicity. Recent temporal and spatially resolved single-cell RNA sequencing (sc-RNA seq) studies have further revealed importance of coordinated and zonal response via various liver cell types in orchestrating liver regeneration and maintenance of essential liver function following APAP hepatotoxicity, which need further exploration [44,45].

## 6. Factors to Consider while Designing Studies to Investigate Liver Regeneration Following APAP Hepatotoxicity

One of the most important factors to consider while studying liver regeneration following APAP hepatotoxicity is that the extent of liver regeneration response is dependent on the amount of initial liver injury [9]. Thus, any interventions altering initial liver injury can indirectly affect liver regeneration secondary to altered injury making it difficult to delineate a direct role of any intervention/mediator in liver regeneration. If the focus of a study is to investigate a direct role of a mediator on liver regeneration, intervention should be done at a late time point such that liver injury is already established. For instance, in our previous study, early treatment with the EGFR inhibitor drastically decreased initial liver injury making it unfeasible to investigate a direct role of EGFR in liver regeneration using this experimental strategy. Therefore, we utilized a strategy of late treatment with EGFR inhibitor (after liver injury was fully established), so that we can study a direct role of EGFR in liver regeneration [8]. Late interventions are especially relevant clinically as NAC is already available as a gold-standard therapy that is highly effective upon early intervention. If late intervention is not possible due to experimental constraints (such as utilization of

transgenic mice model), the effect on liver injury should also be fully characterized, and any alteration of liver regeneration indirectly due to effect on liver injury should be considered. For example, in our previous study, investigation of a direct role of ILK or β-catenin in liver regeneration was hampered using the KO mice strategy as deletion of these genes decreased expression of Cyp2e1 and thus decreased metabolic activation and initial liver injury after APAP overdose [14,37]. Further, it is also important to study the complete time course of liver regeneration phase as some of the interventions/mediators might only produce temporal effects such as delay or early onset of liver regeneration without altering the peak of liver regeneration or the final outcome. For instance, in our previous study, GSK3 inhibition resulted in early initiation of regeneration response following APAP hepatotoxicity, but analysis of complete time-course of regeneration phase revealed that peak regenerative response, overall recovery, and survival were not affected [28]. For studies that are solely focused on investigating the effect of any intervention on the APAP hepatotoxicity phase, the inclusion of regeneration phase time points should also be considered to rule out any deleterious effects of that intervention on liver regeneration and overall recovery. For example, in our previous study, EGFR inhibition showed striking protection against liver injury after APAP overdose in mice; however, it impaired regeneration upon late intervention during the regeneration phase, diminishing the scope of its potential therapeutic utility [8]. Further, as discussed previously, it is critical to consider the use of multiple doses of APAP in a study, considering characteristics and mechanisms of liver injury/regeneration greatly vary in a dose-dependent manner. Especially if the aim of a study is to find interventions to stimulate liver regeneration, higher doses of APAP should be considered. This is because, at moderately toxic doses of APAP, the liver undergoes robust spontaneous regeneration, and only at severe APAP overdose is liver regeneration impaired, reflecting clinical APAP-induced ALF, which requires transplantation. While designing studies using high doses of APAP (such as 600 mg/kg) in mice, it is important to properly control the extent of fasting prior to APAP administration as excessive fasting (more than 12 h) may result in high mortality such that analysis of late time points might be difficult. Lastly, for complete analysis of liver regeneration in APAP-induced ALF model, multiple proliferative markers should be investigated encompassing all the phases of the cell cycle from initiation of cell cycle to mitosis. Further, hepatocyte proliferation parameters should be corroborated other parameters such as regression of injury, final recovery, and survival to demonstrate clinical significance of the interventions on final outcome.

## 7. Concluding Remarks

Timely liver regeneration is an important determinant of final recovery after acetaminophen hepatotoxicity. Liver regeneration after APAP overdose is orchestrated by various signaling pathways activated by myriad of cytokines, growth factors, and endobiotics. Effective communication between hepatic parenchymal and non-parenchymal cells is crucial for regenerative response following APAP-induced liver injury. A summary of the process of liver regeneration and recovery following acetaminophen (APAP)-hepatotoxicity is presented in Figure 1. Liver regeneration after APAP overdose is a dose-dependent compensatory response. The mechanisms involved in regulating regeneration vary greatly with the dose of APAP with overt activation of anti-proliferative signaling pathways after severe APAP overdose. It is important to consider dose–response characteristics of liver regeneration while designing regenerative studies using the APAP model. There is still a need to fully understand the mechanisms that regulate liver regeneration, especially after severe APAP overdose, which is associated with failed spontaneous recovery and adverse outcome. Since liver regeneration can be targeted even after liver injury has been fully developed, regenerative therapies can be especially promising for late presenting APAP-induced ALF patients.

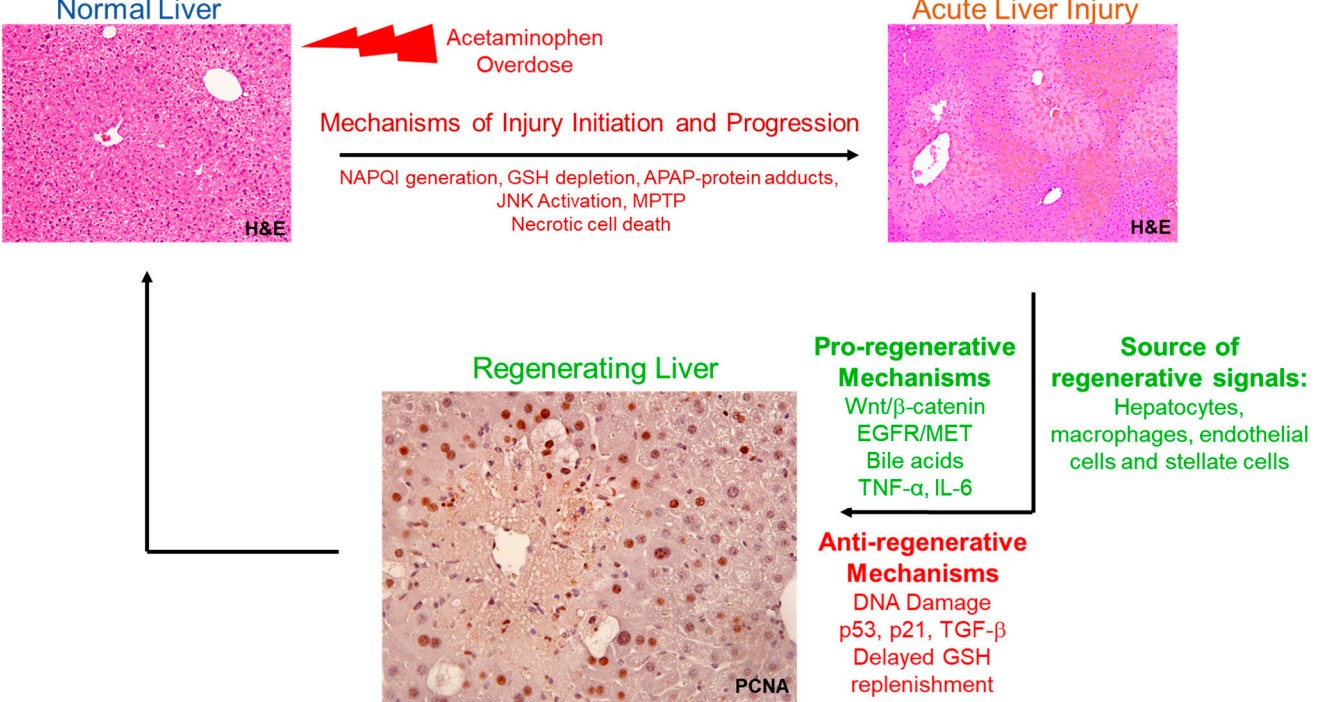

**Figure 1.** Schematics showing an overview of the process of liver regeneration and recovery following acetaminophen (APAP)-hepatotoxicity: APAP overdose initiates a cascade of events resulting in centrilobular liver necrosis. Pro-regenerative signals from various hepatic cell types stimulate hepatocyte proliferation and liver regeneration resulting in regression of liver injury and spontaneous recovery after moderate APAP overdose. Various anti-regenerative signals are also activated for balanced proliferative response and effective repair. However, overt activation of anti-proliferative pathways after severe APAP overdose results in impaired liver regeneration and failed recovery. GSH, glutathione; MPTP, mitochondrial permeability transition pore; NAPQI, N-acetyl-p-benzoquinone imine. Normal liver (**top**, **left**) and necrotic liver after acetaminophen-induced acute liver injury (**top**, **right**) are represented by respective hematoxylin and eosin (H&E)-stained liver sections; Regenerating liver (bottom) is represented by liver section with proliferating cell nuclear antigen (PCNA) positive hepatocytes (brown staining).

**Author Contributions:** All authors contributed equally. All authors have read and agreed to the published version of the manuscript.

**Funding:** Supported by NIH R01 DK135566; NIH R01 DK98414; NIH R01 DK122990; Additional support provided by NIH grant P30 DK120531 to Pittsburgh Liver Research Center (PLRC).

**Institutional Review Board Statement:** Not applicable.

**Informed Consent Statement:** Not applicable.

**Data Availability Statement:** Not applicable.

**Conflicts of Interest:** The authors declare no conflict of interest.

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
