# Peer review of "Regeneration and Recovery after Acetaminophen Hepatotoxicity"

_livers, doi:10.3390/livers3020021_

Round 1

Reviewer 1 Report

The review submitted by expert authors in the field was informative and applicable for general readers. Here are some suggestions.

1.       Cyclin D1 and p21 level are key determinants of fate of injured liver to enter into regeneration phase through Wnt/b-Catenin and TGFb1 signaling pathway respectively, as discovered and described in their original papers. Including those key words in abstracts will be more visible to general readers.

2.        “Multiple signaling mediators contribute to orchestrate liver regeneration following….”. If authors aim to express the “pro-regenerative” factors in this section, it will be the best to includes explicitly review on authors’ original paper  Am J Pathol, 2014. 184 (11): p. 3013 25, and links various regenerative signaling pathways mentioned in the section.

3.       “Signaling mechanisms involved in inhibiting liver regeneration following APAP….” Major advance in this pathway is discovery of role of TGFb1-p21 (Sci Transl Med, 2018. 10 (454). Start with TGFb1 and discuss explicitly on TGFb1-p21 role in regeneration. Remember that TGFb1 inhibitor is potential therapeutic application for regeneration.   

4.       Figure needs one correction. “Anti-regenerative mechanisms ……..” on the left of Regenerative liver micrograph should be on the right side below “pro-regenerative mechanism…….”.

Author Response

We have incorporated all four suggestions of the reviewer in track-change mode in the revised manuscript. 

Reviewer 2 Report

The authors provide an extensive review on delineating liver regeneration mechanisms in acetaminophen-induced liver injury.

The review structure is well organized. The introduction and the conclusion are reasonable, given the article's premise. However, there are following suggestions for the authors to address

  1. Introduction: The closing paragraph needs additional text indicating what various themes in the review are addressed.

  2. Figure 1:The authors have shown the Histopathology and IHC images but have not indicated what the stains represent. The authors should indicate the staining information in the Figure or Figure legend.

  3. The authors should include a scale bar for the IHC images in Figure 1.

Author Response

Thank you for your suggestions. Please see below our response highlighted in yellow: 

  1. Introduction: The closing paragraph needs additional text indicating what various themes in the review are addressed. We have added additional text in the revised version.

  2. Figure 1:The authors have shown the Histopathology and IHC images but have not indicated what the stains represent. The authors should indicate the staining information in the Figure or Figure legend. We have now included stain information both in figure and figure legend for more clarity.

  3. The authors should include a scale bar for the IHC images in Figure 1. Figure 1 is simply a schematic representation of the main point i.e. APAP overdose results in injury followed by liver regeneration and that regeneration will lead to restoration of structure and function. The pictures included are purely representative and not meant to be taken as scientific data. We have added additional text in figure legend for more clarity.

Reviewer 3 Report

The authors describe interesting evaluations and practical aspects of studies aimed at investigating the effect of APAP on liver regeneration.

I suggest some additional points for completeness and to improve relevance:

1. Intro (around lines 82-87) authors highlight that regeneration may differ in a toxic environment (as with APAp intoxication) but this is also true in the context of other associated pathologies like fatty liver or alcohol misuse, which may have an impact on liver regeneration already. A comment on this should be added.

2. The title of the 2nd chapter does not reflect the content, as there are only a couple of statements about doses. It may be appropriate to add more info and relative references (not only the studies from the authors) on the doses usually used in models. Also, I would comment on the relevance of the doses in models when compared to patients (lines 120-125), what is the dose usually found in patients? Would it be more relevant to use animal models with "impaired" liver regeneration in combination with low doses of APAP if this is what happens in real life?

3. in the last chapter starting at line 288: it would be very informative and contribute to the clarity of the chapter to add examples with citations of the mentioned studies focussing on diverse regenerative stages.

Minor comments:

Review the grammar or if there are missing words in the following lines: 100-102;  262; 306.

Many of the cited publications come from the authors themselves, I would recommend increasing the contribution of others to support or improve the discussion of the highlighted points.

Author Response

Thank you for the suggestions. We have incorporated all the suggestions of the reviewer by adding more details in the respective sections (in track change mode) and also corrected the typos.